# Implementation of Next Generation Sequencing-Based Liquid Biopsy for Clinical Molecular Diagnostics in Non-Small Cell Lung Cancer (NSCLC) Patients

**DOI:** 10.3390/diagnostics11081468

**Published:** 2021-08-13

**Authors:** Elisabetta Zulato, Valeria Tosello, Giorgia Nardo, Laura Bonanno, Paola Del Bianco, Stefano Indraccolo

**Affiliations:** 1Immunology and Molecular Oncology Unit, Istituto Oncologico Veneto IOV IRCCS, I-35128 Padova, Italy; elisabetta.zulato@iov.veneto.it (E.Z.); valeria.tosello@iov.veneto.it (V.T.); giorgia.nardo@iov.veneto.it (G.N.); 2Medical Oncology 2, Istituto Oncologico Veneto IOV IRCSS, I-35128 Padova, Italy; laura.bonanno@iov.veneto.it; 3Clinical Research Unit, Istituto Oncologico Veneto IOV IRCCS, I-35128 Padova, Italy; paola.delbianco@iov.veneto.it; 4Department of Surgery, Oncology and Gastroenterology, Università degli Studi di Padova, I-35128 Padova, Italy

**Keywords:** NGS, liquid biopsy, cfDNA, NSCLC, diagnostic routine

## Abstract

Genetic screening of somatic mutations in circulating free DNA (cfDNA) opens up new opportunities for personalized medicine. In this study, we aim to illustrate the implementation of NGS-based liquid biopsy in clinical practice for the detection of somatic alterations in selected genes. Our work is particularly relevant for the diagnosis and treatment of NSCLC. Beginning in 2020, we implemented the use of Roche’s Avenio ctDNA expanded panel in our diagnostic routine. In this study, we retrospectively review NGS-based clinical genetic tests performed in our laboratory, focusing on key analytical parameters. Avenio ctDNA kits demonstrated 100% sensitivity in detecting single nucleotide variants (SNVs) at >0.5% variant allele frequency (VAF), and high consistency in reproducibility. Since 2020, we performed cfDNA genotyping test in 86 NSCLC patients, and we successfully sequenced 96.5% (83/86) of samples. We observed consistency in sequencing performance based upon sequencing depth and on-target rate. At least one gene variant was identified in 52 samples (63%), and one or more actionable variants were detected in 21 out of 83 (25%) of analysed patients. We demonstrated the feasibility of implementing an NGS-based liquid biopsy assay for routine genetic characterization of metastatic NSCLC patients.

## 1. Introduction

Recent advances in molecular biology and cancer genomics improved knowledge of the molecular landscape of cancer, highlighting its heterogeneity and, in parallel, introducing the era of targeted therapies and personalized medicine [1]. The implementation of personalized treatment represents an important turning point in cancer management, and has significantly improved patient clinical outcome.

On the other hand, under the pressure of targeted therapies, cancer cells may select for specific somatic mutations that confer drug resistance, allowing them to survive and drive cancer progression [2]. In this context, early assessment of somatic mutations acquired during disease progression became crucial to offer specific therapeutic options for newly activated signalling pathways or for escape mechanisms that require alternative specific inhibitors.

The necessity of repeated analysis in patient’s tissue specimens is challenging and very often not feasible for clinical reasons.

Liquid biopsy and, specifically, the genetic characterization of circulating free DNA (cfDNA) has emerged as a non-invasive method for the detection of somatic mutations for research, diagnosis, and prognosis of solid tumors. cfDNA analysis can potentially identify actionable alterations in tumor-derived DNA and capture intra-tumor heterogeneity [3].

The role of liquid biopsy for treatment selection of advanced or metastatic non-small-cell-lung cancer (NSCLC) patients is increasing. Due to intrinsic limitations of tissue genotyping, liquid biopsy is currently recommended for NSCLC patients for two applications: detection of EGFR sensitising mutation at diagnosis, when tissue biopsy is not informative, and detection of EGFR resistance mutations at relapse after treatment with EGFR first- or second-generation TKIs inhibitors [3]. However, the surge of new approved therapies targeting specific oncogene-addicted lung cancers is rapidly expanding clinical applicability of cfDNA genotyping [4,5].

Although the detection of somatic gene variants in cfDNA represents a challenge due to the minimal amount of circulating tumor DNA (ctDNA) diluted in cfDNA background released from non-transformed cells, several studies have highlighted the relevance of next-generation sequencing (NGS) approach to detect somatic mutations for the early screening and detection of cancer, and to monitor tumor treatment response and minimal residual diseases [5,6,7,8,9]. Moreover, several NGS commercial platforms to enrich genomic regions of interest for specific cancer type are now available, allowing ctDNA gene sequencing for clinical application.

Recently, we introduced NGS-based liquid biopsy for the detection of somatic alterations in our clinical diagnostic practice, by exploiting a commercial panel including single nucleotide variants (SNVs), fusions, copy number variants (CNVs) and indel mutations (InDels) of selected genes particularly relevant for diagnosis and treatment of advanced NSCLC.

The aim of this study is to retrospectively review NGS-based tests performed in our laboratory in order to illustrate the implementation of NGS based liquid biopsy for clinical molecular diagnostics. We evaluated several key parameters including sequencing performance, the sensitivity, the inter-run variability, and the sequencing coverage.

Our analysis uncovers the potential of NGS for genetic characterization of cfDNA in advanced NSCLC patients.

## 2. Materials and Methods

### 2.1. Patients and Plasma Sample Collection

This is a retrospective review of the sequencing tests performed at our institution, from January 2020 to March 2021. NGS-based liquid biopsy analysis was requested due to the unavailability of tumor tissue for molecular characterization.

Eligibility criteria were confirmed histologically diagnosis of advanced lung cancer.

No specific patients’ informed consent was requested for the present study since no individual patient data were collected nor reported.

Twenty ml of peripheral blood were collected in two Helix cfDNA Stabilization tubes (Streck Corporate, La Vista, NE, USA) and processed within 24–72 h, as previously described [10]. Briefly, blood sample was centrifuged at 2000× *g* for 10 min at 4 °C and the supernatant was subsequently centrifuged at 20,000× *g* for 10 min. Plasma samples were stored at −80 °C until the analysis.

### 2.2. Extraction of cfDNA and Assessment of Its Quantity and Quality

Circulating free DNA (cfDNA) was extracted from 2 to 5 mL of plasma using the AVENIO cfDNA Isolation Kit (Roche Diagnostics, Basel, Switzerland), according to manufacturer’s instructions, and eluted into 60 μL of buffer. cfDNA quantity was assessed using the QuBit dsDNA HS Assay kit with QuBit 3.0 fluorometer (Thermo Fisher Scientific, Waltham, MA, USA). Quality of cfDNA samples was determined with the Agilent 4200 TapeStation using a Cell-free DNA ScreenTape Assay (Agilent Technologies, Santa Clara, CA, USA). The assay evaluates cfDNA samples against a preset region from 50 to 700 bp, including the cfDNA multimer fragments and excluding the high molecular weight DNA.

### 2.3. cfDNA Sequencing with the AVENIO ctDNA Expanded Kit

Sequencing libraries were prepared from 10 to 50 ng cfDNA, using the AVENIO ctDNA Expanded kit (77 genes; Roche Diagnostics, Basilea, CHE, according to the manufacturer’s instructions. Briefly, samples of cfDNA were end-repaired, A-tailed, and ligated with barcoded adaptors. Adaptor ligated samples were amplified by 12 cycles of PCR.

The adaptor-ligated libraries were hybridized for 16–18 h and enrichment of target genes was performed using streptavidin-conjugated magnetic beads (Dynabeads M-270 Streptavidin, ThermoFisher Scientific, Waltham, MA, USA) at 47 °C. Target-enriched libraries were amplified by 15 cycles of PCR and were size-selected for an average fragment size of 300 bp.

Individual enriched libraries were quantified with QuBit dsDNA HS Assay kit, and their profile was assessed using the Agilent 4200 TapeStation using the Agilent High Sensitivity D1000 ScreenTape Assay.

Four or eight purified libraries per run were pooled and sequenced on an Illumina NextSeq 500 (Illumina, San Diego, CA, USA), using the 300-cycle NextSeq 500/550 Mid Output v2 kit or the 300-cycle NextSeq High Output kit, respectively, in paired-end mode (2 × 151 cycles).

### 2.4. NGS Data Analysis

Following sequencing, alignment and gene variant calling were performed using the AVENIO ctDNA analysis software (Roche Diagnostics), with default parameter settings for the expanded panel. The analysis software includes three default reports that are automatically generated: a sample metrics report, an initial variant report (unfiltered or all variants), and a second variant report (Roche default filter). The percentage of aligned reads to the human genome that are within the targeted region (unique depth) according to the manufacturer’s instructions should be >40%. Similarly, the expected median unique depth across bases in the targeted region should be at least 2500×, given 50 ng input cfDNA.

All variants were manually inspected and gene variants present in population databases (EXAC, dbSNP, 1000 genomes) were not considered as relevant. To investigate pathogenicity value, the target variants were submitted to the disease-associated databases COSMIC, VARSOME, and Onco KB, and only variants annotated as pathogenic or likely pathogenic were taken into account.

The level of actionability of specific alterations was evaluated with the ESMO Scale of clinical actionability of molecular targets (ESCAT) [11].

### 2.5. Inter-Run Assay Reproducibility of cfDNA NGS Analysis

The inter-run assay reproducibly of the NGS analysis was assessed by sequencing of cfDNA from different plasma samples across two independent multiplexing sequencing runs to compare their variant allele fraction. In this set of experiments, plasma samples collected according to the spontaneous prospective study called MAGIC-1 [10] were selected. The Ethics Committee of our Institution evaluated and approved the MAGIC-1 study design and informed consent (2016/82, on 12 December 2016). Written informed consent was obtained from all patients before study entry and the study was performed in accordance with the Declaration of Helsinki.

### 2.6. Tissue Genetic Analysis with the AVENIO Tumor Tissue Expanded Kit

Concordance of NGS analyses performed in plasma and tissue was evaluated through the analysis of matched tumor tissue and liquid biopsy samples of patients enrolled in the prospective trial MAGIC-1 [10].

Formalin fixed paraffin embedded (FFPE) tissue samples were processed and analysed using the AVENIO Tumor Tissue Expanded kit, according to the manufacturer’s instructions, which contain matched panel content with the corresponding Avenio ctDNA Expanded assay.

Briefly, DNA was isolated from two 10-μm FFPE sections, and quality of the DNA and optimal input for sequencing library preparation was assessed by qPCR. Extracted DNA was then polished to reduce FFPE-introduced sequencing artefacts and fragmented for library preparation. The libraries were pooled and sequenced on the Illumina NextSeq500 in high output mode using paired-end sequencing (2 × 151 cycles).

### 2.7. Statistical Analysis and Data Analysis

A de-identified dataset was reviewed, and the following information was considered for this study: sequencing performance, number of gene variants identified per sample, and diagnostic findings. All identified variants defined as pathogenic, likely pathogenic, and variants of undefined significance (VUS) according to Varsome were considered, whereas likely benign or benign variants and synonymous variants were not included for statistic analysis. Only somatic alterations with variant allele fraction (VAF) ≥ 0.5% were considered for the following analysis.

Statistical analysis between groups was performed using the Anova Test.

## 3. Results

### 3.1. Panel Overview

Since 2020, we implemented in our clinical diagnostic practice the use of Roche’s Avenio ctDNA Expanded panel.

The Avenio ctDNA Expanded kit consists in a 77-panel genes containing 17 cancer driver genes in U.S. National Comprehensive Cancer Network (NCCN) guidelines and additional 60 frequently mutated genes mainly selected for NSCLC and colorectal cancer. The panel exploits unique molecular indexes (UMIs) to tag individual fragments in order to provide a better quantification of copy number and in silico error suppression, and it is capable of detecting four mutation classes: SNVs, fusions, CNVs, and InDels.

The panel kit includes also a cfDNA extraction kit, with affinity columns.

### 3.2. Technical Assessment of cfDNA NGS Analysis

#### 3.2.1. Assessment of Sensitivity and Limit of Detection (LOD)

To ascertain the limit of detection (LOD) of the method of detecting frequently occurring somatic mutations at low frequency, we first assessed its technical performance using the Multiplex I cfDNA Reference Standard Set and the Structural Multiplex cfDNA Reference Standard (Horizon Discovery, Waterbeach, UK), consisting of genetic alterations of all main classes (SNVs, indels, CNVs, and fusions) tested by the assay.

Specifically, the Multiplex I cfDNA reference standard set comprises three cfDNA reference standards, which include eight well-known DNA variants within the targeted regions of the Avenio expanded panel, with an expected variant allele frequencies (VAF) of 0.1%, 1%, and 5%. Moreover, the set comprises a 100% Multiplex I Wild Type cfDNA reference standard. The 0.1% cfDNA reference standard was sequenced twice, to assess the intra-run repeatability. The 1% cfDNA reference standard was diluted 1:1 with the 100% Multiplex I Wild Type cfDNA reference standard to obtain one sample with an expected VAF of 0.5%. This sample was included to evaluate the sensitivity of this analysis at a VAF centered to 0.5%, which is the limit set by the manufacturer. Finally, in order to assess the detection of fusion genes and CNVs, we also included the sequencing of the Structural Multiplex cfDNA Reference Standard, which consists of seven well characterized mutations, with most of them centered at 5% VAF, RET, and ROS1 fusion variants, and a MET focal amplification.

All libraries were prepared with 50 ng of cfDNA Reference Standard and sequenced using the 300-cycle NextSeq 500/550 Mid Output v2 kit. Results are presented in Table 1. All DNA variants with an expected VAF ≥ 0.5% were correctly detected with high accuracy of VAF estimation. DNA variants with an expected VAF < 0.5% were partially detected (62.5%): four out of eight DNA variants (50%) included in the 0.1% cfDNA reference standard were correctly detected in the first replicate, whereas in the second replicate six out of eight variants (75%) were identified. Short deletions in MET, BRCA2, FBXW7, and FLT3 genes, which are included in the structural multiplex cfDNA reference standard, were not identified, likely due to the design of the panel.

#### 3.2.2. Inter-Run Assay Reproducibility of cfDNA NGS Analysis

In order to evaluate inter-run reproducibility of the cfDNA NGS assay, cfDNA from six different plasma samples were sequenced in two independent multiplexing sequencing runs to compare variant detection and their VAF. We selected plasma samples of patients enrolled in the previously reported prospective study MAGIC-1 [10]. Overall, 23 different gene alterations with VAF over 0.5% were analyzed. Results are reported in Table 2. Twenty-three out of twenty-three (100%) mutations were correctly detected in both the independent multiplexing sequencing runs, with minimal variation in mean VAF. High consistency and linear across technical replicates (*n* = 2) was observed (Figure 1), and the maximum variability observed was 17.2% in the two independent experiments (Table 2).

#### 3.2.3. Comparison of Plasma versus Tissue NGS Analysis

To evaluate the concordance and accuracy of ctDNA detection, we analysed matched tumor tissue and liquid biopsy samples in a group of 21 lung cancer patients enrolled in the previously reported prospective trial MAGIC-1 [10].

All liquid biopsy samples were successfully sequenced and analysed (21/21, 100%). Tumor tissue sample was available for 16 out of 21 patients. Overall, four tissue samples resulted not adequate for DNA extraction (percentage of tumor cells <20%), whereas DNA extracted from other four presented quality and quantity not suitable for NGS analysis. Three samples failed sequencing (poor metric parameters). We successfully sequenced three samples (3/21, 14%) and results are reported in the Table 3. Overall, eight out of the nine gene variants identified in tumor tissue samples were identified in the matched liquid biopsy (89%). Only one variant in the RB1 and one in the MSH6 genes identified in tissue specimens at low frequency (<5%) were not identified in liquid biopsy samples, and, interestingly one SNV in MTOR gene was identified in liquid biopsy and not in the matched tumor tissue sample (Table 3).

### 3.3. Study Population

From January 2020 to March 2021, we performed cfDNA genotyping in 94 plasma samples. Overall, 86 out of 94 sample analyzed (91.4%) were derived from patients with NSCLC and were evaluated for this study. NSCLC samples were collected both at diagnosis, before the start of systemic treatment, in order to search alterations in target genes, and at progressive disease, with the aim to identify gene alterations associated with resistance to first line treatment.

### 3.4. Data Collection and Interpretation

#### 3.4.1. Assessment of ctDNA Quantity and Quality

Almost all samples were extracted from >4 mL of plasma, as indicated by the manufacturer’s instructions. Only one case had low amount of plasma and was successfully extracted from 2.5 mL of plasma.

cfDNA concentrations ranged from 1.5 to 223.75 ng per mL of plasma, with an average of 20.16 ng per mL of plasma. Median size of cfDNA extracted was 265 bp, which is in line with the reported cfDNA size distribution [12]. As assessed by agilent tapestation, the mean percentage of cfDNA was 83% of all DNA extracted. Overall, 84 out of 86 samples showed quality and quantity suitable for NGS analysis (success rate: 97%). Two samples presented less than 10 ng of cfDNA, and were not used for libraries preparation.

#### 3.4.2. Sequencing Performance

On average, 36 million reads were generated from each sample, and the number of reads ranged between 19 and 65 million. Only one sample failed sequencing due to inappropriate metric parameters.

We thus proceeded with 83 samples to examine sequencing metrics depending on the cfDNA amount analyzed per each sample.

The mean sequencing depth across all experiments was 13,478, and was similar in different samples independently from the ng of loaded cfDNA used for library preparation (Figure 2a). Consistently, the on-target rate, defined as the percentage of reads mapped to the target region, was not significantly affected by the cfDNA input (Figure 2c). The median value was 69.7%.

In contrast, the mean depth of coverage with unique reads significantly increased according to the input of cfDNA analyzed (*p* < 0.0001). In particular, when libraries were prepared from 31–40 ng or >40 ng input cfDNA, the unique depth increased significantly compared with lower quantities input (*p* < 0.05 and *p* < 0.001 respectively). In fact, mean depth of unique coverage was similar for 10–20 and 21–30 ng of input cfDNA (3463 and 3883, respectively) and progressively increased for 31–40 and >40 ng of input cfDNA used (5102 and 5884, respectively) (Figure 2b).

#### 3.4.3. Gene Variant Detection

At least one somatic alteration was identified in 63% (52/83) plasma samples, with an average of 1.56 variants per sample (range 1–6). The most frequent altered genes were TP53 (31.5%), EGFR (13%), KRAS (10%), STK11 (4.6%), KEAP1 (3%), PIK3CA, CTNNB1, and MET (2.3%).

Interestingly, the number of alterations detected in plasma samples was not influenced by the amout of cfDNA input used for library generation or by the amount of cfDNA isolated from each plasma sample (Figure 3).

Among all alterations found (*n* = 131), including SNVs with VAF > 0.5%, actionable variants represented 20% of the total (26/131). Actionable alterations were defined according to the tiers proposed by the European Society for Medical Oncology (ESMO). The ESMO scale for clinical actionability of molecular targets (ESCAT) proposed four levels of actionability (I-IV), where I represents the maximum level and IV the minimal level of actionability. Actionable alterations clustered in eight genes: (Figure 4a) including EGFR (12/26, 46.1% of all somatic mutation identified), KRAS (4/26, 15.3%), ERBB2(2/26, 7.6%), PIK3CA (2/26, 7.6%), BRCA1 (2/26, 7.6%), ALK (2/26, 7.6%), MET, and RET (1/26, 3.85). For the majority of patients, actionable mutations were classified as SNVs. For ERBB2 gene we found both an amplification and a pathogenic mutation (S310F). Only one MET amplification and one RET-KIFT5 were found in two different patients, while two ALK fusions (ALK-EML4) were detected (Figure 4a). Finally, all alterations were classified according to ESCAT levels for the NSCLC showing that the majority belong to the I level (14 out of 26) and the remaining were divided between level II (7 out of 26) and level III (5 out of 26) (Figure 4b). The VAF of actionable somatic alteration ranged from 0.5% to 18.9%. Additional actionable alterations with VAF ranged from 0.1% and 0.5% were identified in nine patients.

## 4. Discussion

One of the most promising clinical applications of the cfDNA analysis is the non-invasive tumor genotyping. The possibility to identify oncogenic driver mutations in cfDNA opens new opportunities to guide treatment decision with the final aim to improve the outcome of patients with advanced cancers. Specifically, the application of NGS for cfDNA molecular screening has the potential to improve patient molecular stratification, bypassing safety and feasibility aspects associated with the traditional tumor tissue biopsy and potentially mirroring tumor heterogeneity in time and space.

Currently, several recommendations for the application of NGS sequencing in the clinical setting are available [13]. However, there is a lack of consensus guidelines about technical validation of NGS tests for routine diagnostic.

In this study, we present real-world data on the implementation of the NGS-based liquid biopsy approach for clinical molecular diagnostic routine of NSCLC patients.

The panel employed is the Avenio ctDNA expanded kit from Roche. This commercial multiple-gene panel is optimized for cfDNA application and focuses on actionable hotspot mutations, which have therapeutic as well prognostic values for different solid tumors. The panel employes a target enrichment technology that permit to interrogate in a single workflow all four mutation classes: SNV, indel, and key rearrangements and copy number change.

Moreover, the Avenio ctDNA Expanded panel provides an end-to-end solution from cfDNA extraction to bioinformatics analysis, boosting its application in small laboratories without the need for a bioinformatics team.

We first evaluated the assay across several validation parameters including limit of detection, reproducility, and the sequencing performance, with the main aim to determine its applicability for routine clinical testing. The panel is commercially claimed to be applicable for the identification of gene variants in cfDNA which have often very low VAF. Our technical assessment by using the Multiplex I cfDNA Reference Standard set confirmed 100% sensitivity in the detection of SNV with expected VAF > 0.5%. Instead, sensitivity of SNV with expected VAF < 0.5% did not rank 100%, and therefore, the LOD for the detection of SNV has been set at 0.5% VAF.

It is relevant to note that for the clinical relevant SNVs with expected VAF range from 0.1% to 0.5% the sensitivity was 62.5%. These results are consistent with a recent study using the same panel, in which a 50% sensitivity in detecting SNVs at 0.1% VAF was observed [14]. Although under the LOD, these clinical relevant variations may be important to monitor and to assess logitudinally over time. In this context, recent reports including a substantial percentage of patients with an initially negative test for EGFR exon 20 p.T790M detection will become positive at subsequent testing [15,16], probably due to higher tumor burden or increased tumor shedding capacity, confirmed the concept that repeating a liquid biopsy test might increase its overall sensitivity providing information potentially useful for clinical decision making.

As part of the analytical validation and assay robustness, we assessed the inter-run reproducibility of the specific variant calling, and we observed consistency and high reproducibility in the detection of gene variants and their VAF.

Finally, we evaluated the concordance and accuracy of cfDNA detection with matched tumor tissue samples. For this analysis we could include only a limited number of samples since our diagnostic requests are mainly characterized by non-adequate or insufficient corresponding tumor tissue, reason why a liquid biopsy remains the only option for these patients with the final purpose of identifying specific target therapies. Although the limited number of cases, our findings highlighted the potential of liquid biopsy as alternative strategy to tumor tissue molecular characterization. However, a research oriented study to better dissect the correspondence between tumor tissues and the matched liquid biopsy samples is necessary and is ongoing in our lab for selected samples using the same NGS panel (manuscript under preparation).

Finally, we evaluated the concordance and accuracy of ctDNA detection with matched tumor tissue samples. Although limited to a small number of patients, our findings highlight the potential of liquid biopsy as an alternative strategy to tumor tissue molecular characterization.

We implemented the NGS-based liquid biopsy test in our clinical routing in the first quarter of 2020, and we observed a rapid increase in the number of requests and in the sample volume. In addition to the technical validation, we revised the sequencing performance of 83 NGS-based clinical genetic tests analyzed at our laboratory in 2020, encompassing critical aspects of the complete the NGS testing workflow (Figure 5).

Taking into consideration that ctDNA represents only a small fraction of cfDNA, making it highly challenging to accurately detect gene variants at low VAF from low ctDNA inputs, an appealing feature of the Avenio ctDNA expanded kit is the requirement of a small staring input DNA mass. Among all received plasma samples, only two did not result suitable for the NGS library preparation, resulting in a 97.7% success rate.

The small panel size (192 kb), compared with other commercial panels, allowed sequencing to a higher depth. We observed a very high on-target rate (~70%) and a median unique depth (~6000) using >40 ng input, similarly as described by Verma and collegues [14]. Interestingly, although the depth of coverage with unique sequence reads resulted negatively affected by the cfDNA input, a high median unique depth of ~3500× was observed also using the minimal cfDNA input allowed (10 ng).

We successfully sequenced 96.5% (83/86) samples. At least one gene variant was identified in 52 out 83 NGS-based test (63%), and 30 of them presented pathogenic or likely pathogenic variants. Interestingly, we identified actionable gene variants in 25% of analyzed patients. Our findings are in line with a study recently reported by E. Papadopoulou. et al. [17], in which NGS liquid biopsy analysis was performed in 121 NSCLC patients, using the oncomine lung cell-free total nucleic acid research assay from Thermo Fisher Scientific. At least one mutation was identified in 49% of analyzed samples, and in 14.88% of patients a mutation related to an approved treatment for NSCLC was detected [17].

One possible caviat of cfDNA analysis consists in the detection of false positive mutations as previously described [18]. Several studies which analyzed healthy individuals found alterations in specific genes, mainly associated to clonal hematopiesis [19,20,21]. This issue is particularly important in the context of early cancer detection where these mutations may impact the final analysis, differently from our study, where patients have high tumor loads due to advanced lung cancer diluting eventual non-cancer related hot spot mutations.

Among the limitations of the NGS assay employed in our study, we underline the longer turnaround time (three days) compared with other hybridization-based [22] or amplicon-based enrichment panels. or amplicon-based enrichment panels. On the other hand, the cfDNA genetic testing workflow can be easily integrated into the laboratory’s routine, and the sequencing performance results were advantageous in the detection of somatic mutations in relevant cancer genes with high sensitivity and reproducibility. In addition, variant reports provided by the Avenio cfDNA analysis software need to be checked by an expert molecular biologist, in order to include clinically relevant gene variations which are occasionally not annotated in the database interrogated by the software. Moreover, the call of CNVs needs to be supported by manual supervision of raw data, and it is currently limited to three genes (EGFR, MET, ERBB2).

An important feature not adressed by our study is represented by the lack of clinical correlates, which may be helpful in the interpretation of non informative cases. This is a critical aspect to be investigated in order to reduce the percentage of non informative samples.

Our validation strategy indicates the reproducibility and sensitivity of the selected NGS test. This study underlines the clinical utility of the implementation of the NGS-base liquid biopsy for cfDNA screening of tumor somatic mutations.

## Figures and Tables

**Figure 1 diagnostics-11-01468-f001:**
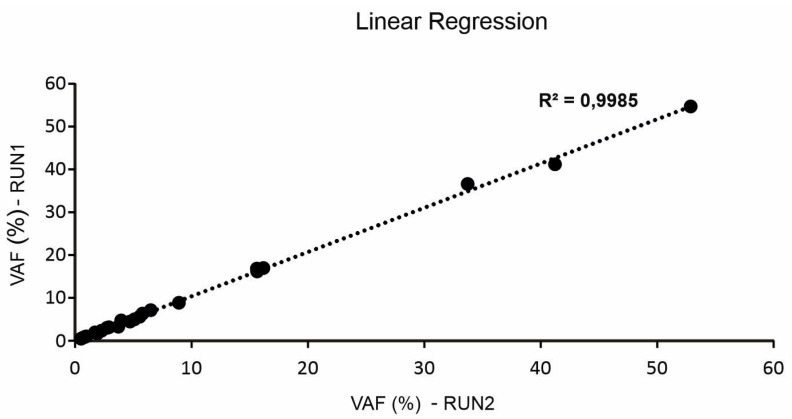
Correlation (o linear regression) of variant allele frequency (VAF) of 23 different gene alterations sequencing in two independent multiplexing sequencing runs.

**Figure 2 diagnostics-11-01468-f002:**
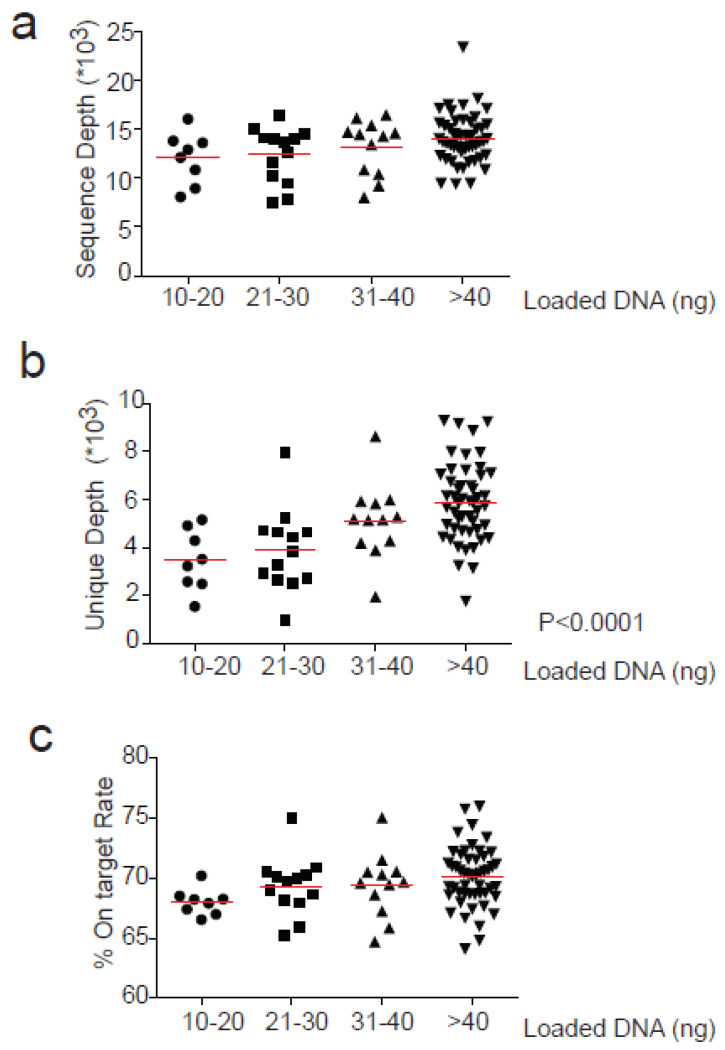
Correlation between loaded cfDNA for library preparation and sequencing parameters. Loaded cfDNA (ng) is divided into 4 groups depending on the quantity used for library preparation (10–20, 21–30, 31–40 and >40) and correlated with (**a**) Sequence depth (**b**) Unique Depth and (**c**) % On-Target rate. Anova test (one-way) was used for statistical analysis for groups comparison. Significant values are reported when *p* < 0.05.

**Figure 3 diagnostics-11-01468-f003:**
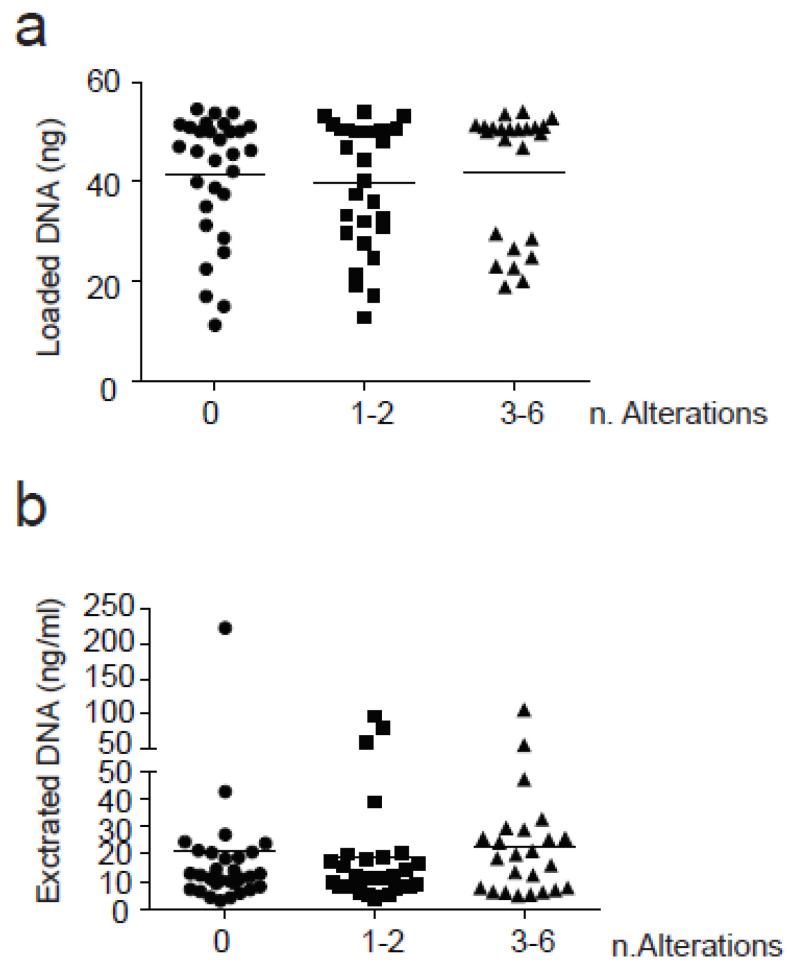
Correlation between the number of alterations found and cfDNA. Three groups based on the number of alterations found (0, 1–2 and 3–6) were correlated with (**a**) Loaded cfDNA(ng) and (**b**) Extracted cfDNA (ng/mL plasma). SNVs were counted with a VAF > 0.5%. Anova test (one-way) was used for statistical analysis for groups comparison. No significant results were found in this analysis.

**Figure 4 diagnostics-11-01468-f004:**
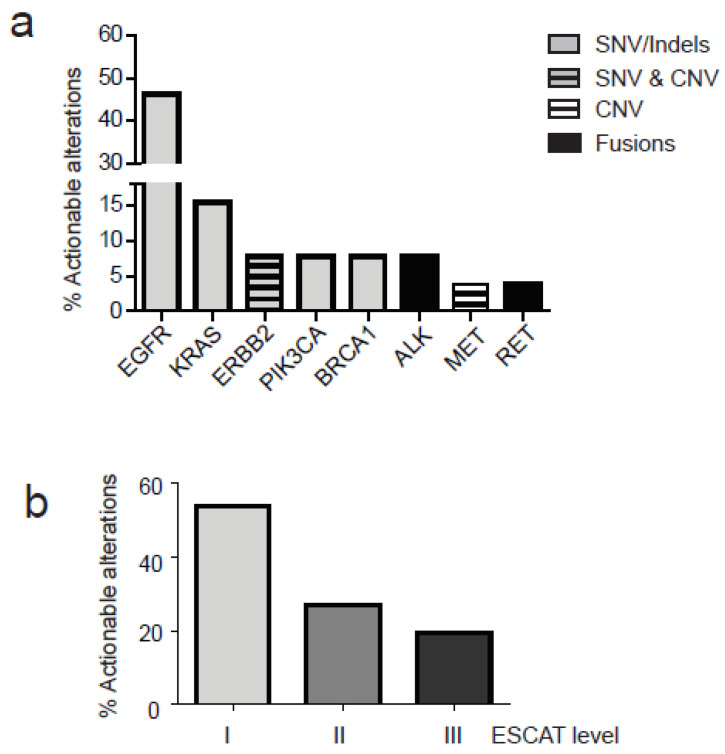
Actionable alterations in NCSCL patients as detected in cfDNA. Actionable alterations with VAF > 0.5% were classified according to ESCAT levels (I, II and III). (**a**) Percentage of actionable alterations in specific genes is represented; different alterations are reported according to the legend (SNVs, Indels, CNV and fusions) (**b**) Actionable alterations are distributed in different levels of actionability according to ESCAT.

**Figure 5 diagnostics-11-01468-f005:**
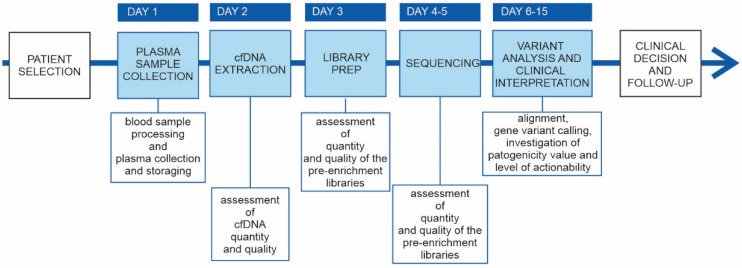
cfDNA genetic testing workflow. Main phases of the cfDNA genetic testing workflow. Phases integrated in the laboratory’s routine are reported in the sky blue boxes; in the white boxs under the timeline are reportered the key aspects for the specific step of the workflow.

**Table 1 diagnostics-11-01468-t001:** Detection of known cfDNA variants at expected VAF in horizon multiplex cfDNA reference standards.

0.1% Multiplex I cfDNA Reference Standard (Part No.: HD779)						
Gene	Variant	Expected VAF (%)	AVENIO_1 Replicate		AVENIO_2 Replicate	
			Detected	VAF	Detected	VAF
*EGFR*	p.L858R	0.10%	NO		YES	0.09%
*EGFR*	p.ΔE746-A750	0.10%	YES	0.11%	NO	
*EGFR*	p.T790M	0.10%	NO		YES	0.12%
*EGFR*	p.V769-D770insASV	0.10%	NO		YES	0.12%
*KRAS*	p.G12D	0.13%	YES	0.14%	YES	0.16%
*NRAS*	p.Q61K	0.13%	YES	0.26%	YES	0.13%
*NRAS*	p.A59T	0.13%	YES	0.19%	NO	
*PIK3CA*	p.E545K	0.13%	NO		YES	0.13%
**Sample with an Expected VAF of 0.5%, Diluted from 1% Multiplex I cfDNA Reference Standard (Part No.: HD778)**						
**Gene**	**Variant**	**Expected VAF (%)**	**AVENIO Results**			
			**Detected**	**VAF**		
*EGFR*	p.L858R	0.50%	YES	0.49%		
*EGFR*	p.ΔE746-A750	0.50%	YES	0.42%		
*EGFR*	p.T790M	0.50%	YES	0.30%		
*EGFR*	p.V769-D770insASV	0.50%	YES	0.29%		
*KRAS*	p.G12D	0.65%	YES	0.40%		
*NRAS*	p.Q61K	0.65%	YES	0.71%		
*NRAS*	p.A59T	0.65%	YES	0.72%		
*PIK3CA*	p.E545K	0.65%	YES	0.43%		
**HORIZON Structural Multiplex cfDNA Reference Standard**						
**Gene**	**Variant**	**Expected VAF (%)**	**AVENIO Results**			
			**Detected**	**VAF**		
*GNA11*	p.Q209L	5.6	YES	4.17%		
*AKT1*	p.E17K	5.0	YES	3.49%		
*PIK3CA*	p.E545K	5.6	YES	4.63%		
*EGFR*	p.V769_D770insASV	5.6	YES	4.50%		
*EGFR*	p.∆E746-A750	5.3	YES	4.70%		
*ROS1*	SLC34A2/ROS1 fusion	5.6	YES			
*RET*	CCDC6/RET fusion	5.0	YES			
*MET*	amplification	4.5 copies	YES			
*KRAS*	p.G13D	5.6	YES	4.41%		
*MET*	p.V237fs	2.5	DEL NOT COVERED BY THE PANEL			
*FLT3*	p.S985fs	5.6	DEL NOT COVERED BY THE PANEL			
*BRCA2*	p.A1689fs	5.6	DEL NOT COVERED BY THE PANEL			
*FBXW7*	p.G667fs	5.6	DEL NOT COVERED BY THE PANEL			
*EGFR*	p.G719S	5.3	YES	5.07%		
*BRAF*	p.V600E	18.2	YES	15.58%		
*PIK3CA*	p.H1047R	16.7	YES	15.73%		
*MYC-N*	amplification	9.5 copies	GENE NOT INCLUDED IN THE PANEL			
*NOTCH1*	p.P668S	5.0	GENE NOT INCLUDED IN THE PANEL			

**Table 2 diagnostics-11-01468-t002:** Evaluation of the inter-run reproducibility assay of the cfDNA NGS analysis in two independent multiplexing sequencing runs.

Sample	Ng Inpu		Gene	SNV	VAF (%)		Mean	St dev
	RUN 1	RUN 2			RUN 1	RUN 2		
1	50	50	*KRAS*	p.Gly12Ala	52.88	54.63	53.76	1.24
			*TP53*	p.Arg306 *	41.22	41.18	41.20	0.03
			*FLT1*	p.Glu201Asp	0.86	0.90	0.88	0.03
			*MSH2*	p.Arg621Leu	16.17	16.95	16.56	0.55
			*MTOR*	p.Ala1792Pro	0.70	0.74	0.72	0.03
			*PTCH1*	p.Val1386Ile	0.50	0.48	0.49	0.01
2	50	50	*KRAS*	p.Gly12Ala	8.91	8.81	8.86	0.07
			*TP53*	p.Arg306 *	5.79	6.31	6.05	0.37
			*MSH2*	p.Arg621Leu	2.89	3.11	3.00	0.16
3	50	27	*TP53*	p.Gln331 *	1.92	1.72	1.82	0.14
			*TP53*	p.Arg248Gln	0.95	0.95	0.95	0.00
			*APC*	p.Gln1611Glu	1.70	1.89	1.80	0.13
4	17,5	23	*BRAF*	p.Gly596Arg	5.53	5.54	5.54	0.01
			*TP53*	p.Gly245Val	3.71	3.26	3.49	0.32
			*DDR2*	p.Glu583Lys	2.26	2.34	2.30	0.06
			*KEAP1*	p.Arg536Pro	4.71	4.43	4.57	0.20
			*TSC2*	p.Ser740Cys	3.94	4.68	4.31	0.52
5	50	20	*RB1*	p.Arg262Gln	15.61	16.80	16.21	0.84
			*TP53*	p.His179Arg	33.71	36.57	35.14	2.02
			*KDR*	p.Ser968Ile	2.72	3.00	2.86	0.20
			*MSH6*	p.Leu1264Val	15.64	16.15	15.90	0.36
			*TERT*	c.-146C > T	6.50	7.07	6.79	0.40
6	50	50	*TP53*	c.993 + 1G > A	5.09	4.98	5.04	0.08

St dev = Standard Deviation; * = (asterisk) = translation termination (stop) codon.

**Table 3 diagnostics-11-01468-t003:** Comparison of plasma versus tissue NGS analysis.

Sample	Gene	SNV	AVENIO cfDNA Expanded Assay	AVENIO Tumor Tissue Expanded Assay
1	*PTCH1*	p.Ala793Ser	detected	detected
	*SMO*	p.Asp25Gly	detected	detected
	*MLH1*	c.1558 + 3A>	detected	detected
2	*TP53*	p.Ile255Phe	detected	detected
	*RB1*	p.Arg251 *	not detected	detected
3	*KRAS*	p.Gly12Ala	detected	detected
	*TP53*	p.Arg306 *	detected	detected
	*MSH2*	p.Arg621Leu	detected	detected
	*MTOR*	p.Ala1792Pro	detected	not detected
	*MSH6*	p.Trp55Arg	not detected	detected

* = (asterisk) = translation termination (stop) codon.

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
