# Peer review of "Implementation of Next Generation Sequencing-Based Liquid Biopsy for Clinical Molecular Diagnostics in Non-Small Cell Lung Cancer (NSCLC) Patients"

_diagnostics, 2021, doi:10.3390/diagnostics11081468_

Round 1

Reviewer 1 Report

The manuscript entitled "Implementation of Next Generation Sequencing-Based Liquid Biopsy for Clinical Molecular Diagnostics in Non-Small Cell Lung Cancer (NSCLC) Patients" highlighted the feasibility of implementing an NGS-based liquid biopsy assay for routine genetic characterization of metastatic NSCLC patients.

  • The Authors should provide the expand forms for alla cronyms, including gene acronyms, through the text when they furst appear.
  • Gene acronyms should be written in italics.
  • In the Methods section the Authors should better describe the pre-analytic phase (sample collection, centrifugation, and in general the handling of liquid biopsy samples) and the run metric parameters adopted to call each recognised variant.
  • Mutations should be reported as follow: e.g. EGFR exon 20 p.T790M.

Author Response

We thank the Reviewer for his/her comment, and we revised the manuscript accordingly. Specifically, we provided the expand forms for all acronyms, gene acronyms has been written in italics and gene variants were reported as suggested. Finally, additional information concerning the pre-analytic phases and run metric parameters has been reported in the Methods section.

Reviewer 2 Report

The submitted work is interesting and it is in the focus of interest.

Mayor comments:

It would be interesting to see and compare the obtained mutation screening results with tissue sample data. It would increase the value of the present work if we would be able to see the difference.

There is no healthy control group involved in the study. It also could increasy the value of the study if we can see that what is the incidence among those who do not have cancer of these mutations or polymorphisms.

Minor comments:

During the cfDNA isolation the range of the obtained DNA concentration varies, what did you get and how could you overcome on it during the sequencing. You you used different quantity of the starting plasma volume, how could it has effect on the results?

You state in the Introduction that you introduced a screening method for liquid biopsy samples which is suitable for detection mutations, deletions, insertions, etc., while you are using a factory developed kit in this work.

Somehow is seems you did not inform the patients about there result, while it is important for them to get the right treatment.

What about the Ethical permission for the study, why you did not apply for that?

You used a Roche kit in your work, the company should already perform similar experiments to be able to put on the market their kit. If you could perform the missing data what I asked among the Mayor comments, you give extra valuable information regarding the applicability of liquid biopsy and the detectable genetic alterations by this ,method in tissues and controls.

Author Response

It would be interesting to see and compare the obtained mutation screening results with tissue sample data. It would increase the value of the present work if we would be able to see the difference.

We agree with the scientific relevance of analysing tumor tissue and compare its molecular profile with results obtained in liquid biopsy.  However, tissue availability often represents a strong limitation to molecular characterization, particularly in NSCLC patients. Specifically, in this study, we retrospectively reviewed NGS-based liquid biopsy analysis performed in NSCLC patients with unavailable tissue left for molecular profiling. This aspect was remarked in the Method section (paragraph 2.1. Patients and plasma sample collection, line 77-78)

There is no healthy control group involved in the study. It also could increase the value of the study if we can see that what is the incidence among those who do not have cancer of these mutations or polymorphisms.

We thank the reviewer for his/her comments. Although the incidence of gene variants in a healthy control group represents an interesting issue to analyse and investigate, analyses reported in the present report were largely performed for diagnostic purposes. Consequently, we had not the possibility to analyse a healthy control group.

Minor comments:

During the cfDNA isolation the range of the obtained DNA concentration varies, what did you get and how could you overcome on it during the sequencing. You used different quantity of the starting plasma volume, how could it has effect on the results?

Among 86 NSCLC samples analysed and evaluated for this study, almost all samples were extracted from the same volume of plasma (>4 ml), as indicated in the paragraph 3.4.1 (Assessment of ctDNA quantity and quality). Only one case had low amount of plasma (2.5 ml) and was successfully extracted and sequenced. It is reasonable to speculate that the broad concentration range of the isolated cfDNA (from 1.5 to 223.75 ng per ml of plasma) was not due to technical aspects but to biological reasons, such as tumour burden and levels of tumour shedding. The concentration of cfDNA in blood varies significantly and it ranges between 0–5 and >1000 ng/ml in cancer patients (Schwarzenbach H, Hoon DSB, Pantel K. Cell-free nucleic acids as biomarkers in cancer patients. Nat Rev Cancer. 2011;11(6):426–437. doi:10.1038/nrc3066. ). This variability is typically attributed to the tumor burden and tumor localization, and recently it has been hypothesized that the cfDNA level could reflect tumor interactions with the microenvironment or the various metabolic properties of progressing cancer (Diehl F, et al. Detection and quantification of mutations in the plasma of patients with colorectal tumors. Proc Natl Acad Sci. 2005;102(45):16368–16373. doi:10.1073/pnas.0507904102; 45;  Xia L, et al. Statistical analysis of mutant allele frequency level of circulating cell-free DNA and blood cells in healthy individuals. Sci Rep. 2017;7(1):1–7. doi:10.1038/s41598-017-06106-1.).

All sequencing libraries were prepared from 10 to 50 ng cfDNA, according to the manufacturer's instructions. In order to evaluate the impact of cfDNA concentration on the final results, we examined sequencing metrics depending on the cfDNA amount analysed per each sample. This is reported in paragraph 3.4.2. (Sequencing performance). Although the depth of coverage with unique sequence reads resulted negatively affected by the cfDNA input, a high median unique depth of ~3500x was observed also using the minimal cfDNA input allowed (10 ng), that is over the expected median unique depth given 50 ng input cfDNA (2500x), as reported in the Methods sections (paragraph 2.4. NGS Data analysis).    

You state in the Introduction that you introduced a screening method for liquid biopsy samples which is suitable for detection mutations, deletions, insertions, etc., while you are using a factory developed kit in this work.

In this study, we present real-world data on the implementation of a commercial NGS-based liquid biopsy test for clinical molecular diagnostic routine of NSCLC patients. We specified this in the Introduction.

Somehow is seems you did not inform the patients about their result, while it is important for them to get the right treatment.

In this study we retrospectively review NGS-based tests performed in our laboratory for routine diagnostic activity. Clinical report is provided to all patients by their oncologist, who is in charge of treatment decisions also based on the results of the genetic test.

What about the Ethical permission for the study, why you did not apply for that?

For this study, no specific patients’ informed consent was requested since no individual patient data were collected nor reported.

In a set of experiments, samples used to evaluate inter-run reproducibility of the cfDNA NGS assay were selected from plasma samples collected according to the spontaneous prospective study called MAGIC-1. The Ethics Committee of our Institution evaluated and approved MAGIC-1 study design and informed consent (2016/82, on 12th December 2016). Written informed consent was obtained from all patients before study entry. The study was performed in accordance with the Declaration of Helsinki. This point was integrated in the paragraph 2.5 of the Methods section (Inter-run assay reproducibility of cfDNA NGS analysis) (lines 137-140).

You used a Roche kit in your work, the company should already perform similar experiments to be able to put on the market their kit. If you could perform the missing data what I asked among the Mayor comments, you give extra valuable information regarding the applicability of liquid biopsy and the detectable genetic alterations by this , method in tissues and controls.

In the methods section, we added some performance parameters of the kit provided by Roche.

Round 2

Reviewer 2 Report

The authors tried to improve the quality of their submitted manuscript. 

I accept most of their response for my questions, but they did not give appropriate answers for the followings:

  1. What is the frequency of the studied polymorphims or genetic alterations in a normal healthy population. You should collect data related that. at least.
  2. If you have left over tissue samples from histological examinations, you can try to get information at least in a few samples, or you can collect parallel samples from new cases (tissue and blood). It could improve the value of your study, as do not forget you used a commercially available kit, you have to add scientific value to your work.

Author Response

The authors tried to improve the quality of their submitted manuscript. 

I accept most of their response for my questions, but they did not give appropriate answers for the followings:

  1. What is the frequency of the studied polymorphims or genetic alterations in a normal healthy population. You should collect data related that at least.

In the present manuscript, and in particular in the paragraph 3.3.3. (Gene variant detection), only variants defined as pathogenic, likely pathogenic and variant of undefined significance (VUS) according to Varsome and not reported in population allele frequency datasets (ExAC and 1000 genomes), were considered.

As previously reaffirmed, investigation concerning the frequency of gene variants in a healthy control group represents an interesting issue. Moreover, with the increasing adoption of whole-genome, exome, and panel-based genetic testing, the detection of novel and uncharacterized sequence variants has steadily increased and defining their frequency in the healthy population is challenging. In any case, this was not the aim of our study.

Nevertheless, to comply with the reviewer’s request, we are planning to analyse with the same NGS panel cfDNA of a group of 7 healthy subjects (lab staff members). Plasma samples were already collected from these volunteers and cfDNA was successfully extracted from all samples. Sequencing data will be available in 40 days, if this is acceptable for the editor.

  1. If you have left over tissue samples from histological examinations, you can try to get information at least in a few samples, or you can collect parallel samples from new cases (tissue and blood). It could improve the value of your study, as do not forget you used a commercially available kit, you have to add scientific value to your work.

As previously reported, we are not able to analyze tissue samples from the same cohort of patients reported in the present manuscript.

Nevertheless, in line with the Reviewer’s comment, we evaluated matched tumor tissue and liquid biopsy samples in a different group of lung cancer patients by using the same NGS panel.

We show here (only for the Reviewer, because these data are submitted for publication elsewhere) results obtained in 23 matched tumor tissue and liquid biopsy samples. All liquid biopsy samples were successfully sequenced and analyzed (23/23, 100%). Left over tumor tissue sample from routine diagnostic activity was available for 16 out of 23 patients. Overall, 4 tissue samples resulted not adequate for DNA extraction (percentage of tumor cells <20%), whereas DNA extracted from other 4 samples was not adequate for NGS analysis. Three samples failed sequencing (poor metric parameters) and library preparation is ongoing for additional 2 samples. We successfully sequenced 3 samples (3/21, 14%) and results are reported in the table below. Overall, 8 out of 9 gene variants identified in tumor tissue samples were identified in the matched liquid biopsy. Only one variant in the RB1 and one in the MSH6 genes identified in tissue specimens at low frequency (<5%) were not identified in liquid biopsy samples, and, interestingly one SNV was identified in liquid biopsy and not in the matched tumor tissue sample.

Results about the 3 samples successfully sequenced both in plasma and tumor tissue biopsy are reported in the main text (paragraph 3.2.3. Comparison of the NGS-based liquid to the NGS-based tumor tissue biopsy).

Although limited to a small number of patients, these findings confirm some known limitations in tissue molecular characterization and highlight the potential of liquid biopsy as alternative strategy.

Sample

Gene

SNV

AVENIO cfDNA Expanded assay

AVENIO Tumor Tissue Expanded assay

1

PTCH1

 p.Ala793Ser    

detected

detected

SMO

 p.Asp25Gly

detected

detected

MLH1

c.1558+3A>

detected

detected

2

TP53

p.Ile255Phe

detected

detected

RB1

p.Arg251*

not detected

detected

3

KRAS

p.Gly12Ala

detected

detected

TP53

p.Arg306*

detected

detected

MSH2

p.Arg621Leu

detected

detected

MTOR 

p.Ala1792Pro

detected

not detected

MSH6

p.Trp55Arg

not detected

detected

4

RB1

p.Arg262Gln

detected

ne-tumor tissue not adequately represented

TP53

p.His179Arg

detected

KDR

p.Ser968Ile                                                                 

detected

MSH6

p.Leu1264Val       

detected

TERT

c.-146C>T

detected

5

KRAS

p.Ala146Thr

detected

ne-tumor tissue not adequately represented

KEAP1

 p.Gly378  

detected

6

no mutation identified

DNA extraction failed

7

TP53

c.993+1G>A

detected

DNA extraction failed

8

AR

p.Gln58Leu 

detected

Sequencing failed

9

KDR

p.Trp364*

detected

DNA extraction failed

10

KRAS

p.Gly12Val

detected

DNA extraction failed

11

APC

p.Glu1345*

detected

Sequencing failed

STK11

p.His154Leu

detected

12

no mutation identified

ne-tumor tissue not adequately represented

13

AR

p.Gln58Leu

detected

ne-tumor tissue not adequately represented

TP53

p.His214Arg

detected

14

ALK fusions

Sequencing failed

15

PMS2

p.Ala38Val

specimens exhausted for routine diagnostic purposes

RET

p.Gly733Asp

TP53

p.Ala159Pro

TP53

p.Leu130His

APC

p.Ser1503*

16

TP53

p.Arg181Pro

specimens exhausted for routine diagnostic purposes

KIT

p.Tyr95Phe

17

no mutation identified

specimens exhausted for routine diagnostic purposes

18

AR

p.Val478Leu

specimens exhausted for routine diagnostic purposes

KDR

p.Tyr927Asp

PDCD1LG2

c.816+2T>G

19

AR

p.Gln58Leu

specimens exhausted for routine diagnostic purposes

NF2

p.Glu112*

TP53

p.ARG158Leu

BRCA2

p.Thr2350Ser

PIK3R1

p.Ser628Asn

RET

p.Arg79Leu

20

TP53

p.Gln331*

specimens exhausted for routine diagnostic purposes

TP53

p.Arg248Gln

APC

 p.Gln1611Glu

21

KEAP1

p.Glu134*

specimens exhausted for routine diagnostic purposes

22

no mutation identified

sequencing on going

23

no mutation identified

sequencing on going

ne- not evaluable

Round 3

Reviewer 2 Report

It is fine if you plan to involve healthy control samples, but seven cases won't be enough to get statistical power and do not forget about the age match.

So you have this data what I was asking, just you sent it out to other journal. It is the issue of Editorial Office how can they deal with that. May be it will be enough to mention it in the ms, that related data is available in that publication.

Author Response

Response to Reviewer 2

  1. It is fine if you plan to involve healthy control samples, but seven cases won't be enough to get statistical power and do not forget about the age match.

We appreciate the reviewer’s comment. The issue of  cancer associated mutations in healthy individuals is an important topic, in particular in the context of early cancer prevention. In fact, cancer related genes, in particular the TP53 gene, can be detected in healthy subjects without any association with subsequent cancer onset (Alborelli I., et al., Cell Death and Disease, 10, 2019; Fernandez-Cuesta L. et al., EBioMedicine, 10 2016). This is true also for genes related to clonal hematopoiesis, that is particularly relevant in older individuals and mainly involves genes such as TET2, DNMT3A, JAK2 and TP53 (Razavi P. et al., Nature Medicine, 25, 2019; Hu Y. et al., Clinical Cancer Research, 2015). Importantly, in the presence of doubts related to clonal hematopoiesis, analysis of peripheral blood cells can help in excluding false positive results. In addition, mutations associated with clonal hematopoiesis are not therapeutically targeted in the context of lung adenocarcinoma, not affecting the main focus of our analysis. Finally, any mutation possibly associated to a germline condition, due to high VAF levels (more than 40% of VAF), or to clonal hematopoiesis are highlighted in our patient’s diagnostic report according to the informed patient consensus. In conclusion, the issue raised by the reviewer concerning a higher number of controls is correct but is not the focus of this study. Our purpose is not prevention and early cancer detection where false positives may affect results. Our goal is to detect specific targeted alterations in patients that are for the majority characterized by advanced disease and high tumor load. Aware of the importance of this topic we added a few sentences in the discussion to better convey this message (green highlighted).

2.

…..So you have this data what I was asking, just you sent it out to other journal. It is the issue of Editorial Office how can they deal with that. May be it will be enough to mention it in the ms, that related data is available in that publication.

We thank the reviewer for the comment. We are aware of the importance of analyzing matched tumor tissue and liquid biopsy using this NGS panel. For this reason we added a more extensive explanation in the discussion (green highlighted, lines…). 
